# A Natural Degradant of Curcumin, Feruloylacetone Inhibits Cell Proliferation via Inducing Cell Cycle Arrest and a Mitochondrial Apoptotic Pathway in HCT116 Colon Cancer Cells

**DOI:** 10.3390/molecules26164884

**Published:** 2021-08-12

**Authors:** Yu-Ting Chou, Yen-Chun Koh, Kalyanam Nagabhushanam, Chi-Tang Ho, Min-Hsiung Pan

**Affiliations:** 1Department of Food Science, National Taiwan Ocean University, Keelung 20224, Taiwan; tinaevachou@gmail.com; 2Institute of Food Sciences and Technology, National Taiwan University, Taipei 10617, Taiwan; yenkoh0123@gmail.com; 3Sabinsa Corporation, East Windsor, NJ 08520, USA; kalyanam@sabinsa.com; 4Department of Food Science, Rutgers University, New Brunswick, NJ 08901, USA; ctho@sebs.rutgers.edu; 5Department of Medical Research, China Medical University Hospital, China Medical University, Taichung City 40402, Taiwan; 6Department of Health and Nutrition Biotechnology, Asia University, Taichung City 41354, Taiwan

**Keywords:** feruloylacetone, demethoxy-feruloylacetone, colon cancer, curcumin, degradant

## Abstract

Feruloylacetone (FER) is a natural degradant of curcumin after heating, which structurally reserves some functional groups of curcumin. It is not as widely discussed as its original counterpart has been previously; and in this study, its anticancer efficacy is investigated. This study focuses on the suppressive effect of FER on colon cancer, as the efficacious effect of curcumin on this typical cancer type has been well evidenced. In addition, demethoxy-feruloylacetone (DFER) was applied to compare the effect that might be brought on by the structural differences of the methoxy group. It was revealed that both FER and DFER inhibited the proliferation of HCT116 cells, possibly via suppression of the phosphorylated mTOR/STAT3 pathway. Notably, FER could significantly repress both the STAT3 phosphorylation and protein levels. Furthermore, both samples showed capability of arresting HCT116 cells at the G2/M phase via the activation of p53/p21 and the upregulation of cyclin-B. In addition, ROS elevation and changes in mitochondrial membrane potential were revealed, as indicated by p-atm elevation. The apoptotic rate rose to 36.9 and 32.2% after being treated by FER and DFER, respectively. In summary, both compounds exhibited an anticancer effect, and FER showed a greater proapoptotic effect, possibly due to the presence of the methoxy group on the aromatic ring.

## 1. Introduction

Colorectal cancer (CRC) is the third most common cancer, ranked fourth for its fatality rates, and its incidence is increasing year by year. As reviewed in 2017, there were up to 4–5% people suffering from colorectal cancer and the development of CRC could be associated with personal features or habits such as age, chronic disease history, and lifestyle [1]. The common treatment for colon cancer usually involves surgery to remove the cancer, radiation therapy and chemotherapy. However, CRC is often difficult to treat, and is characterized by a high chance of relapse [2].

Curcumin is a bioactive compound extracted from the plant *Curcuma longa*, widely used in traditional medicines due to its antioxidant, anti-inflammatory, antidiabetic, antiviral, antimicrobial [3], anticancer [4], immunoregulatory [5], cardio- [6] and hepatoprotective [7], as well as neuroprotective properties [8,9]. However, curcumin’s applications are limited, due to its low cellular uptake, low chemical stability and low water solubility which result in poor oral bioavailability [10]. Nevertheless, curcumin provides a wide range of derivative compounds, such as tetrahydrocurcumin (THC) [11], dihydrocurcumin (DHC) [12], hexahydrocurcumin (HHC) [13], etc. Due to the similarities between the parts in the functional groups, most of them exhibit eye-catching bioactivity in disease prevention. Moreover, some derivative compounds have overcome the shortcomings of curcumin by ameliorating its drawbacks and enhancing its bioavailability [14]. 

In some studies, it has been confirmed that curcumin possesses an inhibitory ability regarding the proliferation, colony formation, migration and tumorsphere formation of colon cancer cells [15]. Curcumin is well known for both its Michael donor and Michael acceptor units, to the extent that the latter have been suggested in previous studies to contribute to the inhibitory effect on the activation of NF-κB [16,17]. Besides NF-κB, transcription factors such as AP-1 and STAT3, and protein kinases including PKA and PKC, are the molecular targets of curcumin that have previously been revealed [18]. The presence of Michael acceptors allows the occurrence of nucleophilic addition reactions with -OH, -SH and -SeH, which are known as the Michael addition [19]. For instance, glutathione with -SH (thiol) and thioredoxin reductase with -SeH (selenol) of selenocysteine can easily undergo 1,4-addition with curcumin through covalent bonding [19]. 

One of the major characteristics of curcumin is that it is relatively unstable, so may degrade into several products in a neutral to alkaline solution [20]. The degradation products of curcumin that have been identified previously include ferulic acid and feruloylmethane from hydrolysis as the minor degraded products [21], and bicyclopentadione from autoxidation as the major degraded product [20]. Feruloylacetone (FER), also recognized as 6-(p-hydroxy-m-methoxyphenyl)-5-hexane-2,4-dione (half-curcumin), was firstly introduced by Feng and Liu in 2011 through artificial synthesis [22]. In fact, FER is a natural degradant found in the curcumin product when heated to 100 °C, indicating that the transformation from curcumin into FER may frequently occur during household cooking [23]. In fact, diketene curcumin, as another naturally found degradant in curcumin-containing cuisine, was suggested to exhibit anticancerogenic activity against B78H1 melanoma cells, as a result of cell cycle G2 arrest [24]. 

In comparison, the study by Feng and Liu (2011) is the only study that discussed demethoxy-feruloylacetone (DFER) as a dihydroxy-feruloylacetone or 6-(m,p-dihydroxyphenyl)-5-hexene-2,4-dione [22]. Both FER and DFER previously exhibited antioxidation activity, and the results showed that there were faster decrements when scavenging DPPH radicals than galvinoxyl radicals, indicating that FER and DFER might have a higher antioxidation activity than their precursor [22].

As mentioned above, FER might be a major degradant product of curcumin in household cooking, one that is highly likely to be reaching the digestive tract. Therefore, the major aim of our study is to investigate its anticancer ability, primarily focusing on colon cancer. Furthermore, to investigate the effect of phenolic hydroxyl- or methoxy groups on the compound’s efficacy, FER is compared to DFER, its counterpart with a substitution of a hydroxyl instead of a methoxy group on the phenolic ring. 

## 2. Materials and Methods

### 2.1. Compounds, Reagents and Antibodies

Feruloylacetone (FER) and dimethoxy-feruloylacetone (DFER) were kindly provided by Sabinsa Corporation (New Jersey, USA). Both compounds were dissolved in DMSO for experiments because they were compared to curcumin or curcumin-related analogues that were mostly reported to be dissolved in DMSO due to low water solubility. Both FER and DFER could be fully dissolved in DMSO up to the concentration of 30 g/L while they were both not fully dissolved at the concentration of 1 g/L (Data not shown). The primary antibodies mTOR (#7C10), p-mTOR (#2974), PARP (#9542), p-atm (#5883), cyclin B1 (#4138) and β-actin were purchased from Cell Signaling (Beverly, MA, USA). p21 (sc-817) was purchased from Santa Cruz Biotechnology, Inc. (Dallas, TX, USA). p53 (ab131442) and CDK1 (ab18) were purchased from Abcam Co (Cambridge, UK). Caspase 9 (66169-1-lg) was purchased from Proteintech Group, Inc. (Illinois, USA), while caspase 3 (31A1067) was purchased from Novusbiologicals (Colorado, USA). The secondary anti-mouse and anti-rabbit antibodies were purchased from Cell Signaling. The Bio-Rad protein assay dye reagent was purchased from Bio-Rad Laboratories (Munich, Germany) [25].

### 2.2. DPPH Assay

A DPPH assay was used to detect free radical scavenging activity in different concentrations of the samples. Samples of 40 µL of FER at concentrations of 20, 10, 5, 2.5, and 1.25 (mM), or 625, 313, 156, 78, and 39 (µM), or DFER at concentrations of 20, 10, 5, 2.5, or 1.25 (mM), blank and standard, were added to 200 µL of DPPH methanolic solution (Sigma, Burlington, VT, USA) in 96-well plates. After being incubated in the dark for 1 h, the absorbance under 517 nm was read. The absorbance of the control solution was in the range of 0.8 ± 0.1. The free radical scavenging activity was calculated using the following equation:(1)Free radical scavenging activity (%)=[(Acontrol−Ablank)Astandard]×100%.

### 2.3. Cell Culture

Human colon cancer cell line HCT116 was purchased from American Type Culture Collection (ATCC) (Maryland, USA). The cells were maintained in RPMI 1640 medium supplemented with 10% fetal bovine serum and 1% penicillin [26]. The cells were then cultured at 37 °C in a humidified atmosphere containing 5% CO_2_.

### 2.4. Cell Viability Assay

Cell viability was determined using the MTT assay. HCT116 cells were seeded at a density of 4 × 10^5^ cells/mL in a 96-well plate and treated with different compounds at various concentrations for either 24 or 48 h. The medium was removed and washed with PBS once, before incubating the cells with 3-(4,5-dimethylthiazol-2-yl)-2,5-diphenyltetrazolium bromide (MTT, Sigma, Burlington, USA) solution at the final concentration of 0.02% for 45 min. DMSO-dissolved formazan was read at the wavelengths of 570 nm and 690 nm. The cell viability was calculated using the following equation:
(2)Cell viability (%)=[(A570−A690)Acontrol]×100%.

### 2.5. Apoptotic Cell Observation

For the observation of apoptotic cells, the procedure of Smith et al. (2012) was followed [27]. HCT116 cells were seeded and treated with FER and DFER at various concentrations for 24 h. The cells were centrifuged at 1000 rpm for 3 min before being stained with ethidium bromide and acridine orange (EtBr/AO) for 1–2 min at the final concentration of 50 μg/mL. The cells were observed at 488 nm/530 nm and 530 nm/580 nm.

### 2.6. ROS Level Determination by Flow Cytometry

Cells (4 × 10^5^ cells/well) were cultured in 24-well plates and treated with FER or DFER at a concentration of 50 µM. After treatment for 24 h, the cells were collected and washed with phosphate-buffered saline (PSB) twice, and trypsinization was carried out. After centrifugation, the cells were incubated with 2 µL DCHF-DA (Sigma, Burlington, USA) in PBS for 1 h in the dark at 37 °C. The fluorescence intensity was determined by flow cytometry at an excitation wavelength of 488 nm and an emission wavelength of 530 nm.

### 2.7. Determination of Mitochondrial Membrane Potential

HCT116 cells were seeded at the density of 4 × 10^5^ cells/mL for 12 h, followed by sample treatment for 24 h at the concentration of 25 μM. The cells were collected by trypsinization after PBS wash. Centrifugation at 3500 rpm for 10 min followed by PBS wash was conducted twice and the cells were resuspended in PBS before staining. The cells were incubated with either DiOC6 or JC-1 at the final concentration of 1 nM or 2 μM, respectively. The fluorescence intensity was determined by flow cytometry. For DiOC6, the excitation wavelength was 488 nm and the emission wavelength was 530 nm. For JC-1, the excitation wavelength was 488 nm, while the emission wavelengths were 530 nm and 580 nm.

### 2.8. Cell Cycle Analysis

The 12 h-seeded HCT116 cells (4 × 10^5^ cells/well) were treated with FER and DFER (25, 50 µM) for 24 h. The treatment cells were harvested using trypsin-EDTA and washed with PBS. Then, cold 70% ethanol was used to fix the collected cells and they were stored at -20 °C overnight [26]. After centrifugation, the cells were washed with PBS to remove the ethanol. The fixed cells were incubated with hypotonic buffer at 37 °C for 15 min before propidium iodide (PI) (Sigma Chemical Co., St. Louis, MS, USA) staining for another 15 min. Fluorescence intensity by flow cytometry was determined at an excitation wavelength of 488 nm and an emission wavelength of 580 nm.

### 2.9. Apoptosis Assay

Cell apoptosis was determined by employing the Annexin V FITC apoptosis detection kit (Dojindo, München, Germany), according to the procedure given in the technical manual. Cell collection followed the same method as above, and the cells were stained with Annexin V and FITC, followed by incubation for 15 min. The fluorescence intensities were measured under the wavelengths of 480/530 nm for Annexin V and 530/580 nm for FITC.

### 2.10. Western Blot Analysis

HCT116 cells (4 × 10^5^ cells/well) were cultured in 5-cm culture plates and treated with different concentrations (25, 50 µM) of FER and DFER. After treatment, the cells were harvested and lysed with a lysis buffer containing a protease inhibitor cocktail and 1% of EDTA [28]. Protein concentrations were determined using a BCA protein assay kit (Sigma Chemical Co., St. Louis, MS, USA). The protein samples were loaded on 10% sodium dodecyl sulfate polyacrylamide gel electrophoresis (SDS-PAGE), and separated using a constant voltage of 50 V for 30 min and 100 V for another 4 h. After SDS-PAGE, the proteins were transferred onto polyvinylidene difluoride (PVDE) membranes. The membranes were then blocked with a blocking solution for 1 h. Afterward, the membranes were blotted using primary antibodies for mTOR, p-mTOR, STAT3, p-STAT3, p53, p21, p-atm, cyclinB1, CDK1, PARP, caspase 3 and caspase 9. The membranes were washed three times with TPSB (0.05%) before incubating with secondary antibodies for 1 h. The bands were visualized using chemiluminescence (ECL). 

### 2.11. Statistical Analysis

All results are presented as mean ± S.E. All experiments were performed for at least three replications. Student’s *t*-test was employed to determine statistical significance (*p* < 0.05).

## 3. Results

### 3.1. Distinct Functional Groups May Lead to a Difference in the Antioxidant Activity of FER and DFER

The appearance and structural differences between FER and DFER are as presented in Figure 1a,b. To determine their radical scavenging ability, a DPPH assay was employed. The DPPH radical scavenging ability of FER was 93.47%, and of DFER was 25.56%, at a concentration of 5 mM, showing the superior antioxidation activity of FER as compared to DFER. Up to 50% of the DPPH radical scavenging ability was shown at a very low concentration (313 µM) of FER, indicating a greater potential for FER in terms of its antioxidative capability. The IC_50_ of FER and DFER was 0.2421 mM and 12.96 mM, respectively. 

### 3.2. Both FER and DFER Inhibit Cell Proliferation in HCT116 Colon Cancer Cells

In order to understand the inhibitory efficacy of FER and DFER, these compounds were compared to curcumin and those of its analogs that have been well studied. The results showed that FER and DFER had similar cell viability (almost 40%) at 50 µM (Figure 1f), exhibited a better inhibitory efficacy than tetrahydrocurcumin (THC) and hexahydrocurcumin (HHC); however, the major component of curcuminoids (curcumin (CUR), demethoxycurcumin (DMC) and bismethoxycurcumin (BDMC)) had a greater effect in comparison. A similar result was obtained when comparing all the compounds at a concentration of 100 µM (Figure 1e). When further comparing the differences between FER and DFER at concentrations of 25, 50, 75, and 100 µM at 24 and 48 h (Figure 1g,h), little variation in terms of cell viability was observed under the different treatments, but the IC_50_ of FER was slightly lower than DFER, which might be due to the presence of the phenyl methoxy group.

### 3.3. FER and DFER Exhibited an Antiproliferative Effect on HCT116, Possibly via an Inhibitory Effect on the Phosphorylation of the mTOR/STAT3 Pathway

Based on the results of the MTT assay, it was assumed that FER and DFER might exhibit a suppressive effect on HCT116 proliferation. Therefore, Western blotting was employed to investigate the underlying mechanism. The mammalian target of rapamycin (mTOR) is a downstream kinase in the PI3K/Akt pathway, the activation of which is correlated with an increase in PI3K/Akt-dependent Ser2448 phosphorylation, and which regulates cell growth by integrating nutrient- and growth factor-derived signals [29]. In our results, as compared to the control group, p-mTOR decreased in the treated cells, and the treatment of FER at 50 µM showed the most significant decrement (*p* < 0.0001) (Figure 2d,e). Furthermore, phosphorylation of the downstream target protein STAT3 significantly decreased after both treatments. Notably, FER could also reduce the expression of STAT3 at a concentration of 50 µM. Therefore, the results suggested that both FER and DFER could effectively reduce the phosphorylation of mTOR and STAT3, resulting in antiproliferation activity. 

### 3.4. FER and DFER Lead to Different Responses in Cell Cycle Arrest 

Because of the observable suppression of HCT116 proliferation, the cell cycle stages were investigated by employing flow cytometry with PI staining. As presented in Figure 3, HCT116 cells were arrested at the G_2_/M phase after being treated with either 50 μM of FER or DFER, along with a significant reduction in cells at the G_0_/G_1_ phase. Surprisingly, a significant elevation at the G_0_/G_1_ phase was observed when the cells were treated with FER or DFER at a concentration of 25 μM.

To further explore the mechanism involved, the protein levels of p-atm, p21, p53, CDK1, and cyclin B1 were determined. The activation of the p-atm/p53/p21 pathway could be observed with the cells arrested in both G1 and G2 phase [30,31]. As upregulation of p53 was observed, we investigated the expression of cyclin B, as p53 and cyclin B are normally negatively correlated [32]. Besides, cyclin B and CDK1 complex are responsible for regulating the progression of the G2/M phase. The phosphorylation of atm (Figure 4a) was observed as a response to DNA damage in HCT116 treated with FER or DFER at 50 μM, accompanied with a significant increment in p53 and p21 (Figure 4b,c). In addition, the accumulation of cyclin B1 was also observed when the cells were treated at the concentration of 50 μM. On the other hand, HCT116 cells treated with 25 μM of either FER or DFER were found to show an increment in p53 (Figure 4b) with a significant reduction in CDK1 and cyclin B1 (Figure 4d,e). The results showed that FER and DFER might lead to different responses in cell cycle arrest at different concentrations. 

### 3.5. Increment of ROS and Changes in Mitochondrial Membrane Potential Are Induced by FER and DFER 

An elevation in the phosphorylation of atm might imply DNA damage as a result of ROS production. Therefore, the ROS level was further determined by DCFH-DA staining (Figure 5f,g). The results showed that ROS was significantly induced by both FER and DFER. It was suggested that a low level of ROS might increase the sensitivity of cancer cells to the treatment, while a high level of ROS may induce apoptosis via mitochondrial dysfunction [33,34]. Therefore, the mitochondrial membrane potential (MMP) was investigated by using DiOC6 and JC-1 staining. An abnormal change in MMP was observed and JC-1 aggregates significantly reduced as the monomers dramatically increased after treatment. This result indicated that FER and DFER could significantly induce ROS in HCT116 and bring about an adverse effect on mitochondrial membrane potential.

### 3.6. FER and DFER Induce Cell Apoptosis in HCT116 Colon Cancer Cells

Mitochondrial abnormalities may possibly lead to cell apoptosis. Therefore, an ethidium bromide/acridine orange (EtBr/AO) staining assay was employed for cell apoptosis observation. In Figure 6, it can be observed that the morphology of HCT116 changed at a concentration of 50–100 μM, while 25 μM was not as indicative in terms of morphology in DFER but slightly changed in FER. However, it was obvious that green- and orange-stained cells were sharper in all treatment groups, indicating the presence of condensed or fragmented chromatin. Orange-stained cells sharply increased as the sample concentrations elevated. The staining assay indicated the occurrence of apoptosis; therefore, we further confirmed the proapoptotic effect of FER and DFER with a Western blotting assay.

The normal function of poly (ADP-ribose) polymerase-1 (PARP-1) is the routine repair of DNA damage by adding poly (ADP ribose) polymers, in response to a variety of cellular stresses [35]. When PARP-1 is deactivated, this causes DNA fragmentation and induces apoptosis [35]. The Western blot analysis showed that cells treated with FER and DFER significantly reduced the full form of PARP at a concentration of 50 μM. The occurrence of apoptosis was further supported by the increased cleavage of PARP, caspase 3 and caspase 9 (Figure 7). Lastly, PI/Annexin V staining by using flow cytometry was employed to determine the apoptotic efficacy of the samples. 

As presented in Figure 8, treatment with FER and DFER sharply reduced the cell survival rate and induced cell apoptosis, as expected. Notably, there were statistical differences between FER and DFER in terms of the survival rate and apoptosis rate, indicating the importance that might be provided by the methoxy group on the phenolic ring. 

## 4. Discussion

In this study, the heat degraded product of curcumin, feruloylacetone and its dimethoxy counterpart was investigated for its anticancer capacity. It was found that both samples exhibited antiproliferative and antiapoptotic effects via the inactivation of mTOR/STAT3 pathways and ROS-induced mitochondrial dependent apoptosis cascade. Besides, cell cycle arrest at different phases was achieved at different concentrations and the accumulation of cyclin B was surprisingly observed when the cells were arrested at the G2 phase. 

Curcumin is found to be relatively stable as a nutraceutical food and supplemental product, and its poor solubility is its most recognized disadvantage [36]. In fact, it was reported that heating could significantly increase the water solubility of curcumin up to 12-fold, and that heat-solubilized curcumin was also suggested for consideration in clinical trials for better bioavailability [37,38]. Although it was revealed that culinary heating methods (boiling, roasting and frying) could potentially reduce the total antioxidant capacity of curcuminoids, the remaining antioxidative capacities were still sufficient for neuroprotection [39]. As heating has become a potential strategy to improve the poor solubility of curcumin, degradation caused by high temperature could not be avoided. Therefore, in this study, the natural degradant of curcumin was investigated for its bioactivity, focusing on the colon cancer-suppressive effect. 

Both FER and DFER preserve half of the Michael acceptors, or so-called diketone groups, present in curcumin, with or without the methoxy group in the bulky substituent region that remained. In 2015, Aggarwal et al. revealed in their review that THC, the curcumin analog without diketone groups, could be more effective in modulating mTOR, p53 and p21 [40]. Therefore, the involvement of Michael acceptors in modulation should be further determined. Moreover, as pointed out by Giordano and Tommonaro (2019), curcumin could significantly suppress the expression of mTOR and STAT3 and, at the same time, positively regulate the expression of p53/p21 and caspase-3 activation in several types of cancer [41]. Surprisingly, the degradation of curcumin, resulting in the loss of half of the functional parts that might be crucial in cancer prevention, could still reserve a similar modulative effect on the protein involved. However, as presented in Figure 1e,f, the suppressive effect of FER and DFER was not as efficacious as curcumin, DMC and BDMC, indicating that the presence of these functional groups in the structure is essential for their conspicuous cancer-suppressive effect. However, compared to THC and HHC, which lack diketone groups, FER and DFER showed a greater inhibitory effect on cell viability, indicating the importance of the ketone group on cancer prevention. Notably, there was no significant difference between THC and HHC, showing that Michael donors were not as essential without the presence of the diketone groups. It is suggested that these ketone and enol groups may contribute to direct hydrogen bonding with proteins or the nucleophilic attack by thiol and/or selenol on the protein structure [42]. 

In this study, both FER and DFER showed their potential for colon cancer prevention via three strategies, including antiproliferation, cell cycle arrest and inducing apoptosis. For the inhibition of the mTOR/STAT3 pathway, it was previously suggested that curcumin could directly interact with STAT3 at the Cysteine 259 residue due to the presence of α, β-unsaturated carbonyl moiety [43]. It is worth mentioning that THC without the electrophilic moiety showed a lack of binding ability to STAT3 in the same study, which indicates that the ketone group could be essential for direct binding to STAT3. Our findings showed that FER could significantly reduce the expression and phosphorylation of STAT3. Notably, DFER did not show statistical significance in STAT3 reduction. Therefore, the electrophilic group could be important for inhibiting STAT3 activation, but the methoxy group on the aromatic ring may have a degradative effect on STAT3 expression. A reduction in STAT3 phosphorylation may be involved in apoptosis induction or antiproliferation, which should be further confirmed.

HCT116 colon cancer cells were found to be arrested at the G2/M phase after treatment in this study, accompanied by upregulation of the expression of p53, p21 and Cyclin B. It was previously reported that Cyclin B-CDK1 might play a role in inducing mitotic arrest apoptosis [44]. Furthermore, p53 and p21 activation were suggested to correlate with the G2/M phase cell cycle arrest in this study. It was previously noted that p53 participates in arrest in both G1 and G2 [45,46]. In addition, it has also been suggested that p53 and p21 are essential to arrest DNA-damaged cells at the G2 checkpoint [47]. The activation of p21 could lead to an accumulation of CDK1 and cyclin B in the form of an inactive complex [48]. In 2012, Yadav et al. demonstrated that Gatifloxacin could induce cell cycle arrest at the S and G2 phases via the activation of p21 and p53 in the pancreatic cancer cell line, a finding that was similar to our result [49]. Therefore, it is likely that cyclin B accumulation is involved in G0/G1 arrest. As demonstrated by Su et al. (2006), the downregulation of CDK1, Cdc25c and cyclin B1 was induced by curcumin in the human colon cancer cell Colo 205 [50]. Besides, accumulation of cyclin B1 could also be a result of characteristic or prometaphase arrest [51]. Therefore, the role of accumulated cyclin B1 should be further clarified. 

It is noteworthy that FER and DFER seemed to arrest HCT116 at the G0/G1 phase at a lower concentration. However, the result was accompanied by the suppression of CDK1 and cyclin B, which may cause cell cycle arrest at the G2 phase [52]. We suggest that a low concentration of FER and DFER might not be able to prolong the period of HCT116 cell arrest at the G2/M phase without the activation of p21. Therefore, HCT116 cells pass through the mitotic phase and lead to the phenomenon of G0/G1 phase arrest. It was suggested by Sherwood et al. that the cells could undergo cell division under these preferred conditions [53]. The accumulation of cyclin B1 and CDK1 was suggested as being a result of cell cycle transition [54]. 

As abovementioned, Michael acceptors are well known functional groups in the structure of curcumin due to the benefits brought by the Michael addition in the prevention of various diseases [55]. However, the contribution of aromatic rings should not be neglected. The importance of the methoxy group on the aromatic ring has been reported in several studies [56,57]. In 2020, Huang et al. suggested that curcumin and DMC showed a better proapoptotic effect than BDMC in HOS osteosarcoma cells [56]. Furthermore, it was suggested that the phenyl methoxy group may crucially affect direct inhibitory activity on NF-κB [57]. As a direct target of curcumin [58,59], the interaction or binding with STAT3 might also be influenced by the presence of the phenyl methoxy group, besides the effect of the diketone groups. [60] Furthermore, the methoxy group also contributes to DPPH scavenging activity [61], which was in agreement with our assumption and results. In another study, curcumin exhibited a superior capability in reducing protein carbonyl in lead-induced rat hippocampus as compared to DMC and BDMC and notably, BDMC was the weakest compound of all of them [62], which indicated the benefits of the phenyl methoxy group. Therefore, in this study we suggested that the presence of a phenyl methoxy group may play a crucial role resulting in the greater proapoptotic effect of FER, as compared to DFER. 

## 5. Summary

This study concludes that the natural degradant of curcumin possesses a suppressive effect on colon cancer, in terms of antiproliferation, inducing cell cycle arrest and leading to cell apoptosis. The underlying mechanisms involved in anticancer activity were partially clarified in this study, but the physicochemical properties and bioactivity of these degradants should be further investigated. In short, both FER and DFER are suggested as potential compounds to be further analyzed in order to enhance the medicinal value of curcumin, which will be involved in future clinical studies.

## Figures and Tables

**Figure 1 molecules-26-04884-f001:**
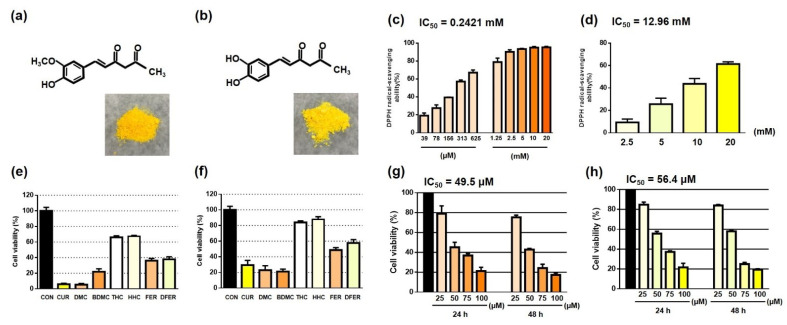
FER and DFER exhibit an antiproliferative effect on colon cancer HCT116 cells. (**a**,**b**) Structural configuration and appearance of FER and DFER. (**c**,**d**) DPPH radical scavenging ability and IC50 of FER and DFER. (**e**,**f**) Cell viability of HCT116 after being treated by curcumin and its analogs or degradants, at the concentration of 100 or 50 μM for 24 h. (**g**,**h**) Cell viability of HCT116 after being treated by FER or DFER at various concentrations for 24 or 48 h.

**Figure 2 molecules-26-04884-f002:**
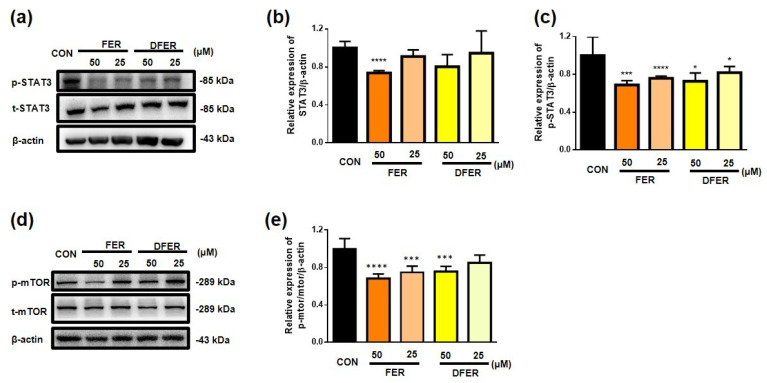
FER and DFER may inhibit cell proliferation by reducing phosphorylation of the mTOR/STAT3 pathway. (**a**) Representative p-STAT3 and STAT3 protein expressions of Western blotting; and (**b**,**c**) quantification of p-STAT3 or STAT3 by Western blotting, using β-actin as an internal control. (**d**) Representative p-mTOR and mTOR protein expressions of Western blotting; and (**e**) quantification of p-mTOR/mTOR by Western blotting, using β-actin as an internal control. Symbols (*), (***) and (****) indicate significant differences (*p* < 0.05), (*p* < 0.005) and (*p* < 0.001), respectively, analyzed using Student’s *t*-test, as compared to the control group.

**Figure 3 molecules-26-04884-f003:**
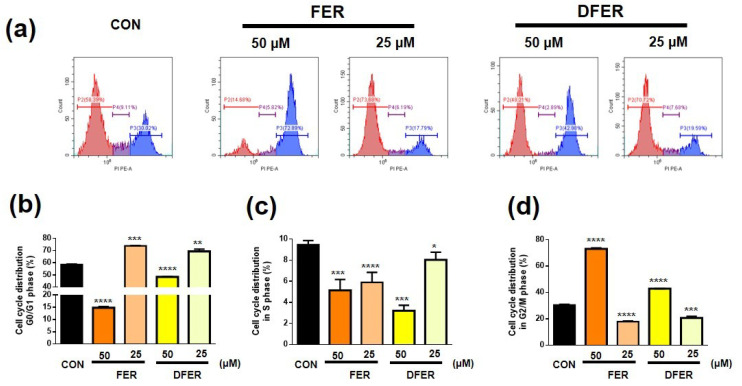
Treatment of FER and DFER arrest of HCT116 at the G2/M phase with a concentration of 50 μM for 24 h. (**a**) Representative cell cycle distribution of FER- and DFER-treated HCT116 at either 25 or 50 μM for 24 h. (**b**–**d**) Percentage of HCT116 distribution at G0/G1 phase, S phase and G2/M phase, respectively. Symbols (*), (**), (***) and (****) indicate significant differences (*p* < 0.05), (*p* < 0.01), (*p* < 0.005) and (*p* < 0.001), respectively, analyzed using Student’s *t*-test, as compared to the control group.

**Figure 4 molecules-26-04884-f004:**
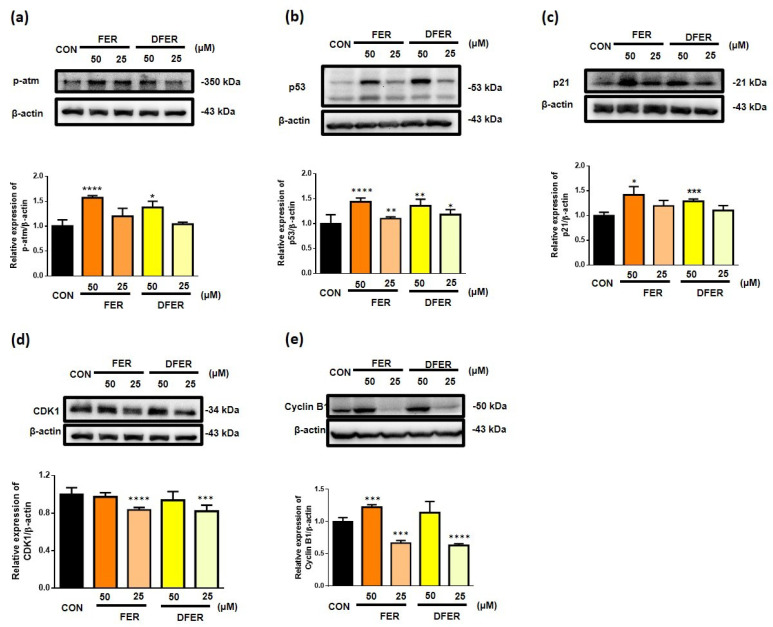
Both FER and DFER may lead to cell cycle arrest by activating the p53/p21 pathway and upregulating cyclin B1 expression. Representative protein expression and quantification of Western blotting of (**a**) p-atm, (**b**) p53, (**c**) p21, (**d**) CDK1 and (**e**) cyclin B1, by using β-actin as an internal control. Symbols (*), (**), (***) and (****) indicate significant differences (*p* < 0.05), (*p* < 0.01), (*p* < 0.005) and (*p* < 0.001), respectively, analyzed using Student’s *t*-test, as compared to the control group.

**Figure 5 molecules-26-04884-f005:**
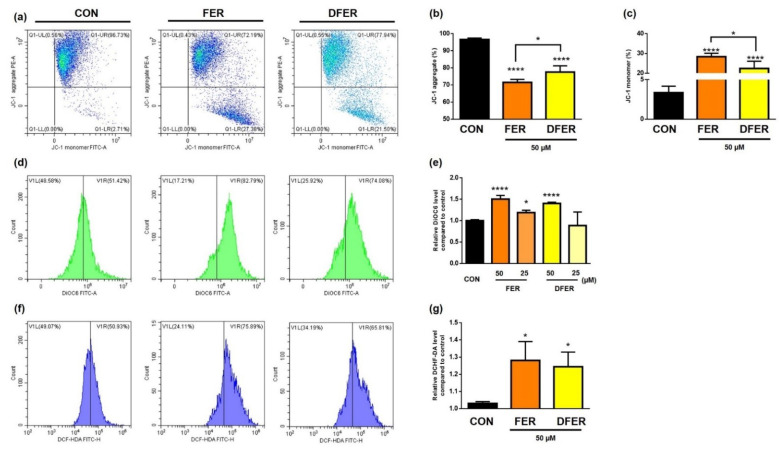
FER and DFER induce ROS and cause MMP abnormality in HCT116 cells. (**a**) JC-1 aggregate and monomer distribution, and relative levels of (**b**) JC-1 aggregate and (**c**) monomer in FER- or DFER-treated HCT116 cells, revealed by JC-1 staining. (**d**) Fluorescence intensity of DiOC6 staining and (**e**) relative level of the intensity in FER- or DFER-treated HCT116 cells by DiOC6 staining. (**f**) The ROS level analyzed by using DCFH-DA staining, and (**g**) relative level. Symbols (*) and (****) indicate significant differences (*p* < 0.05) and (*p* < 0.001), respectively, analyzed using Student’s *t*-test, as compared to the control group.

**Figure 6 molecules-26-04884-f006:**
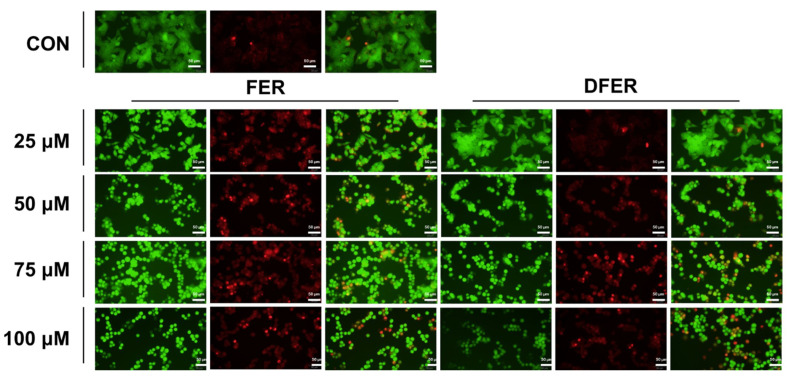
Changes in cell morphology and DNA condensation occur in FER- or DFER-treated HCT116 at different concentrations. HCT116 cells were treated with FER or DFER at various concentrations and were stained with EtBr/Acridine Orange to observe the changes in cell morphology and DNA condensation or fragmentation. Condensed or DNA fragmentation or necrotic cells were stained with observable sharp light green or orange, under excitation/emission wavelengths of 488/530 nm or 530/625 nm, respectively.

**Figure 7 molecules-26-04884-f007:**
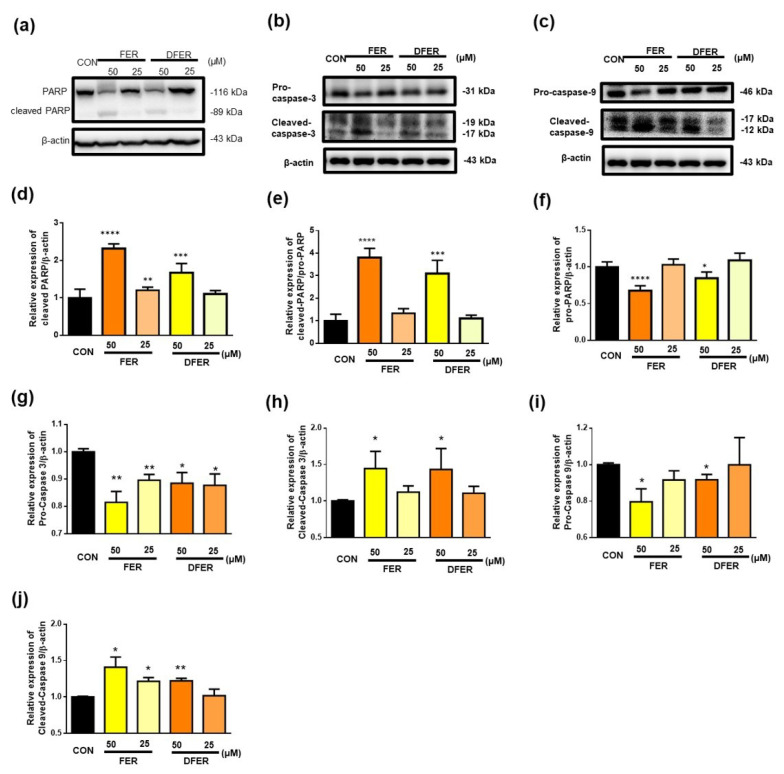
Treatment with FER and DFER for 24 h may induce cell apoptosis via the intrinsic pathway. Representative protein expressions of both (**a**) PARP and cleaved PARP, (**b**) caspase-3 and cleaved caspase-3, (**c**) caspase-9 and cleaved caspase-9, and quantification of Western blotting of (**d**–**f**) PARP (**g**,**h**) caspase-3 and (**i**,**j**) caspase-9 by using β-actin as internal control. Symbols (*), (**), (***) and (****) indicate significant differences (*p* < 0.05), (*p* < 0.01), (*p* < 0.005) and (*p* < 0.001), respectively, analyzed by using Student’s *t*-test, as compared to the control group.

**Figure 8 molecules-26-04884-f008:**
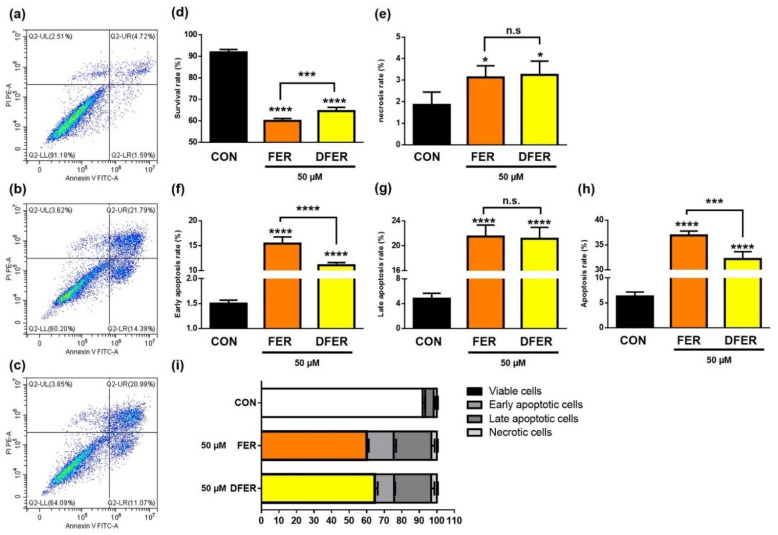
FER and DFER significantly increase the apoptotic rate of HCT116 after 24-h treatment. The percentage of apoptotic or necrotic HCT116 was analyzed by PI/Annexin V staining, and the distributions are as presented in (**a**–**c**). The differences between (**d**) survival rate, (**e**) necrotic cells, apoptotic cells at (**f**) early apoptosis stage, (**g**) late apoptosis stage, and (**h**) overall apoptotic rate are presented. The percentage of each stage is as shown in (**i**). Symbols (*), (***) and (****) indicate significant differences (*p* < 0.05), (*p* < 0.005) and (*p* < 0.001), respectively, analyzed using Student’s *t*-test, as compared to the control group.

## Data Availability

Data available on request due to restrictions e.g., privacy or ethical. The data presented in this study are available on request from the corresponding author.

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
