# Peer review of "A Natural Degradant of Curcumin, Feruloylacetone Inhibits Cell Proliferation via Inducing Cell Cycle Arrest and a Mitochondrial Apoptotic Pathway in HCT116 Colon Cancer Cells"

_molecules, 2021, doi:10.3390/molecules26164884_

Round 1
Reviewer 1 Report
Reviewer comments and suggestions
This study investigated the effect of Feruloylacetone (FER) on colon cancer. In addition, demethoxy feruloylacetone (DFER) was applied to compare the effect which may suggest the structural differences of the methoxy group. The study result showed that both FER and DFER inhibited the proliferation of HCT116 cells, possibly via suppression of the phosphorylated mTOR/STAT3 pathway.
Particularly, FER significantly repressed both the STAT3 phosphorylation and protein levels. Both compounds showed capability of arresting HCT116 cells at the G2/M phase via the activation of p53/p21 and upregulation of cyclin-B. ROS elevation and changes in mitochondrial membrane potential were revealed. The study suggested that these compounds exhibited an anti-cancer effect, and FER has a greater pro-apoptotic effect.
Below are the comments which need to be incorporated in the revised version of the manuscript.
- Line 27 The line needs some modification “in previous studies and, therefore, its anti-cancer efficacy is investigated herein”
- Line 36 have a typo error “rised” please modify it
- Please provide valid reference” from colorectal cancer at present is about 4%–5%, “
- All references cited in the paper were wrong, please modify it based on the MDPI journal
- Line 52-53 One reference is not supported all the properties, please add more references for the aforementioned information
- Line 56-58 need a reference and also for line 397
- Line 61, in some studies need more than 2-3 references otherwise not use the word
- Line 72-73 need to modify the sentence
- Section 2.7 Please explore the step methods used in the study
- Line 246-247 May I know how the author connects these protein or subunits
- Line 269 already mentioned staining, repeating is not good
- Line 299-300 need a valid reference and lines 370-371 also
- In the discussion section Para needed to rewritten as this was a general one, the author needs to report his study finding in first para of the discussion, after that they can discuss other points
- Line 328-330 Confusing sentence
- have the study monitored all “the activation of p21, p27 and p53 in pancreatic cancer cells, which finding was similar to our result” please provide the figure or table
- Please check the correctness of line number 401
- Line 406-408 information was not matched with the above line
- I found lots of mistakes in the reference style of the manuscript and therefore it is advised to the authors to please modify it based on the journal guidelines.
Reviewer 2 Report
In this manuscript, the authors tested the in vitro anti-proliferative effects of both FER and DFER on HCT116 cells, and revealed several related underlying molecular mechanisms. Overall, the study is kind of interesting and has series of decent data. However, I have the following queries. The manuscript can be accepted after addressing the minor revisions noted.
1. Several concerns on data presented in the study:
1) According to Figure 1(e-f), it showed higher cell viability of HCT116 at 100 μM concentration when compared with 50 μM both for FER and DFER. This is not reasonable and should be double checked. Also, when look Figure 1(e) and Figure 1(h) together, the data seems like not consistent for the DFER at 50 μM for 24 h. The Figure 1(e) showed ~40% cell viability, while it was almost ~60% in Figure 1(h). Please also double check this.
2) Regarding to the Figure 4, the corresponding description in Results part was not reasonable. In line 249, the p53 should be p-atm. The sentence “In addition, the accumulation of cyclin B1 was also observed” was not comprehensive since the Figure 4(e) also showed an obvious decrease for the cyclin B1 at the 25 μM concentration. Therefore, please overlook this part and make it more accurate.
2) In terms of the Figure 6, first please add the scale bar. As well, in line 285, the authors described that “25 μM was not as indicative in terms of morphology”, which was not completely right. And it seems pretty easy to see the obvious changes for the FER at 25 μM when compared with CON.
3) In Figure 7, please add Figures alongside with both 7(f) and 7(g) to show the quantitative expressions of cleaved caspase-3 and cleaved caspase-9, respectively.
2. Whether the authors could specify the water solubility of the FER or DFER in the manuscript? As well, whether the authors used DMSO to dissolve the FER or DRER first when prepared the corresponding solutions for the cells culture studies? It would be better to provide this information.
3. The entire manuscript must be carefully edited to enhance the clarity and conciseness and to eliminate grammatical and syntax errors.
Round 2
Reviewer 1 Report
No more comments